# On the Use of Mouse Actions at the Character Level

Ángel Navarro *[ID] and Francisco Casacuberta

Research Center of Pattern Recognition and Human Language Technology, Universitat Politècnica de València, 46022 Valencia, Spain; fcn@prhlt.upv.es
* Correspondence: annamar8@prhlt.upv.es

**Abstract:** Neural Machine Translation (NMT) has improved performance in several tasks up to human parity. However, many companies still use Computer-Assisted Translation (CAT) tools to achieve perfect translation, as well as other tools. Among these tools, we find Interactive-Predictive Neural Machine Translation (IPNMT) systems, whose main feature is facilitating machine–human interactions. In the most conventional systems, the human user fixes a translation error by typing the correct word, sending this feedback to the machine which generates a new translation that satisfies it. In this article, we remove the necessity of typing to correct translations by using the bandit feedback obtained from the cursor position when the user performs a Mouse Action (MA). Our system generates a new translation that fixes the error using only the error position. The user can perform multiple MAs at the same position if the error is not fixed, each of which increases the correction probability. One of the main objectives in the IPNMT field is reducing the required human effort, in order to optimize the translation time. With the proposed technique, an 84% reduction in the number of keystrokes performed can be achieved, while still generating perfect translations. For this reason, we recommend the use of this technique in IPNMT systems.

**Keywords:** Interactive Machine Translation; Mouse Actions; Neural Machine Translation; Computer-Assisted Translation tools; effort reduction

## 1. Introduction

Machine Translation (MT) has improved dramatically over the last few years, thanks to the appearance and application of neural models in the field. With these advances, human parity has been accomplished in several tasks [1], and we can begin to think about a future where MT systems will not need translators to correct their translations, in order to meet the requirements in some domains. Until then, a subfield of investigation in the MT field has focused on the development of new techniques and tools to facilitate the work of professional translators through MT.

Computer-Assisted Translation (CAT) tools [2] include all tools and techniques developed to facilitate human work in the MT field. The main aim of these tools is to generate high-quality translations using the knowledge and experience of professional translators, while reducing their effort. There are different groups of techniques, depending on how they approach the problem. Attempts have been made to reduce human effort through the use of Interactive Machine Translation (IMT) systems, which focus on the interactions between humans and machines. In particular, in Interactive-Predictive Neural Machine Translation (IPNMT) systems [3], each human interaction results in a new prediction from the Neural Machine Translation (NMT) system.

Before the introduction of neural models, a set of ambitious critical projects set milestones in the Interactive-Predictive Machine Translation (IPMT) field. In their time, Transtype2 and CasMacat [4,5] aimed to create workbenches grouping all of the innovative features which were unavailable in other tools, in an attempt to reduce human effort through different approaches, such as intelligent auto-completion [6], confidence measures [7], and word alignment visual information [8]. At present, with the advent of

neural models, research on such workbenches seems to have stopped; however, research on new techniques to reduce human effort in the IPNMT field is still being carried out. Some projects have changed the displays used in the translation sessions to make the process easier and more intuitive [9,10], while others generate additional data that facilitate the translation process [11,12]. Finally, other projects have studied new and minimalist feedback formats to provide to the machine [13,14].

In all of these projects, we find two main IPMT approaches, depending on the order of the corrections performed by the user. When the user always corrects the first error found in a left-to-right manner, a prefix approach is followed [3]; meanwhile, if the user can validate and correct segments anywhere in the translation, a segment approach is used [15,16]. In this project, we use a prefix approach: the user always corrects the first error in the translation. Figure 1 illustrates a conventional IPNMT session. We initially provide a source sentence $x$ to the system, and the user attempts to obtain the reference translation $y$ through different iterations. In the first iteration, the machine produces the hypothesis, $\hat{s}_h$, which includes the error that the user will correct in the next iteration. To correct the error, in the next iteration, the user moves the cursor to the '*submitted*' word position, validating the prefix **p** and typing the correct word $k$. With this new information, the machine produces the suffix $\hat{s}_h$. At this time, the hypothesis is correct, and the user validates it in the second iteration by typing the special token '#'.

| | | SOURCE (x): En 1989 la comisión presentó una nueva propuesta. |
|---|---|---|
| | | REFERENCE (y): In 1989, the commission presented a new proposal. |
| **ITER-0** | (**p**) | ( ) |
| | ($\hat{s}_h$) | *In 1989, the commission submitted a new suggestion.* |
| **ITER-1** | (**p**) | In 1989, the commission |
| | ($s_t$) | *submitted a new suggestion.* |
| | ($k$) | presented |
| | ($\hat{s}_h$) | *a new proposal.* |
| **ITER-2** | ( **p**) | In 1989, the commission presented a new proposal. |
| | ($s_t$) | ( ) |
| | ($k$) | (#) |
| | ($\hat{s}_h$) | ( ) |
| **FINAL** | (**p** ≡ y) | In 1989, the commission presented a new proposal. |

**Figure 1.** Example of a conventional IPNMT session for translating a sentence from Spanish to English. Non-validated hypothesis are displayed in italics, and accepted prefixes are printed in normal font.

IPNMT aims to reduce the effort required of professionals to obtain high-quality translations. In this context, we define the human effort as the cost of performing keystrokes for correction, but we do not consider the cognitive cost of reading the suffix and considering multiple correction options. Considering the example in Figure 1, if the user does not use the IPNMT system, they would need to perform two different corrections; however, with the system, the machine is good enough to produce the correct suffix with only the first modification. In the same way, we can work at the character level, where the user only needs to correct the erroneous first character. Figure 2 illustrates the same example at the character level. In the first iteration, the user only has to type the character '*p*' to provide the system the information about the start of the word. The system can correctly generate the translation in the next iteration with this information. Instead of typing 17 characters to fix the translation, the user only needed to type one. In the same way, Mouse Actions (MAs) can be used to reduce the effort of the user, by trying to eradicate the keyboard actions performed by the user in the IPNMT sessions.

The idea of MAs was initially introduced by Sanchis-Trilles et al., in 2008 [14,17]. The core concept is that a machine can fix translation errors with respect to only their position. This process is effective when the used models are good enough to guess the correct translation on the second or third try. Furthermore, with the improvement of MT

models, it is more probable that the machine will get the translation right on the second try, thus increasing the usefulness of the technique. Sanchis-Trilles et al. have only proved the validity of the technique in conjunction with Statistical Machine Translation (SMT) models in their study. More recently, Navarro et al. (2021) [18] have tested MAs with NMT models at the word level.

In this article, we focus on reducing the human effort in IPNMT systems to decrease the time needed for translation sessions. We use the feedback obtained from the cursor position when the user executes an MA to correct the translation, operating at the character level. In this way, the user can perform MAs in the middle of words, thus validating them partially. By implementing this technique in an IPNMT system, we can reduce the Character Stroke Ratio (CSR) in a translation session by 50% using only one MA, and by 84% with a maximum of five MAs at the same position.

| **SOURCE** (x): | | En 1989 la comisión presentó una nueva propuesta. |
|---|---|---|
| **REFERENCE** (y): | | In 1989, the commission presented a new proposal. |
| **ITER-0** | $(\mathbf{p})$ | ( ) |
| | $(\hat{s}_h)$ | *In 1989, the commission submitted a new suggestion.* |
| **ITER-1** | $(\mathbf{p})$ | In 1989, the commission |
| | $(s_t)$ | *submitted a new suggestion.* |
| | $(k)$ | p |
| | $(\hat{s}_h)$ | *resented a new proposal.* |
| **ITER-2** | $(\mathbf{p})$ | In 1989, the commission presented a new proposal. |
| | $(s_t)$ | ( ) |
| | $(k)$ | (#) |
| | $(\hat{s}_h)$ | ( ) |
| **FINAL** | $(\mathbf{p} \equiv y)$ | In 1989, the commission presented a new proposal. |

**Figure 2.** Example of a conventional IPNMT session at the character level for translating a sentence from Spanish to English. Non-validated hypotheses are displayed in italics, and accepted prefixes are printed in normal font.

## 2. Related Work

The IPMT paradigm was proposed as an alternative to post-editing, and dates back to Foster et al. (1997) [3], who introduced a new interaction style between the translator and the system. First, the translator selects a section of the source text and begins typing its translation. Then, after the user types a character, the system displays a list of possible words that the user may accept or reject. Knowles and Koehn (2016) [19] implemented an IPMT system with NMT models, where the translator corrects the sentence in a left-to-right manner, and the system predicts the best suffix for a validated prefix. NMT models are well-suited for this problem, as they produce the word probabilities for each token from left to right. This procedure describes the conventional behavior of IPNMT systems. Based on this idea, different projects—such as that described in this paper—have attempted to reduce the number of keystrokes performed during interactive translation sessions.

Peris et al. (2017) [16] and Huang et al. (2021) [20] have changed the order in which the user makes the corrections, letting them perform any translation changes without worrying about the position of the error. This kind of IPNMT system is called segment-based.

Some projects have taken advantage of the IPNMT system to improve the MT model used. Peris et al. (2019) [21] have implemented online learning techniques in the system, in order to train the model using the error-free translations corrected by the user. Lam et al. (2019) [22] and Zhao et al. (2020) [23] opted to use reinforcement learning techniques to increase the quality of the translations generated by the MT models, resulting in a reduction in human effort. Both simplified the interaction method by only selecting a set of incorrect words from the translation, or by only checking some critical words, and used the feedback provided by the user not only for correcting the current translation, but also to improve the quality of future translation by using it to train the MT model.

As in this article, some techniques explained below attempt to reduce the associated human effort by directly decreasing the total number of words that the user has to type. One technique that follows this idea is using confidence measures at the word level. Confidence measures provide a quality estimation for each word, using different methods. Some frameworks obtain these confidence measures at the word- or sentence-level [24,25]. Gonzales et al. (2010) [7] used confidence measures to limit the number of words that the user needs to check: all words with a confidence higher than a set threshold do not require revision by the user, such that the final number of words to be corrected by the user is reduced.

With the same idea (i.e., directly reducing the total number of words that the user has to type), Sanchis et al. (2008) [14] have introduced the concept of MA at the word level using SMT models. This concept first appeared with the idea that Interactive-Predictive Statistical Machine Translation (IPSMT) systems could take advantage of the first click that the user makes to perform a correction, where the system tries to fix the translation with just the error position and, in the worst scenario, the user has to enter the words they already have in mind. This work achieved a significant reduction in human effort. More recently, Navarro et al. (2021) [18] have implemented this technique in an IPNMT framework and conducted a comparison of the results, obtaining a higher reduction in human effort. With advances in technology, NMT models generate higher quality translations, thus improving the probability that the system can fix errors without any extra information.

In this work, we extend the idea of MAs in IPNMT systems to the character level. At the word level, the system tries to correct the word with respect to only the error position and, if the correction is still incorrect, the user must type it entirely. However, when working at the character level, the prefix validated by the user can include a few characters from the incorrect word, thus facilitating the correction process; furthermore, if the correction performed is still incorrect, the user only has to type a new character. We assess the improvements brought by this paradigm, due to the change from word to character level in the experimental setup.

## 3. IPNMT Framework

Before explaining the framework implemented for the MAs, we have to look at the NMT framework and how the system generates the translations. Then, we describe how the system incorporates human feedback and takes it into account in the generation process. Castaño and Casacuberta (1997) [26] introduced the NMT framework, and its power has been demonstrated in recent years [27–29]. Given a sentence $x_1^J = x_1, ..., x_J$ from a source language $X$, to obtain the translation with the highest probability $\hat{y}_1^{\hat{I}} = \hat{y}_1, ..., \hat{y}_{\hat{I}}$ from the target language $Y$, the fundamental equation of the statistical approach to NMT is as follows:

$$\hat{y}_1^{\hat{I}} = \arg\max_{I, y_1^I} \Pr(y_1^I \mid x_1^J) = \arg\max_{I, y_1^I} \prod_{i=1}^{I} \Pr(y_i \mid y_1^{i-1}, x_1^J), \qquad (1)$$

where $\Pr(y_i \mid y_1^{i-1}, x_1^J)$ is the probability distribution of the next word, given the source sentence and the previous words. The IPNMT framework is characterized by its use of human feedback as helpful information to calculate the translation with the highest probability. At the word level, when the professional user locates an error following a left-to-right review in the translation generated by the system, they move the cursor to the error position $p$, producing the feedback $f_1^p = f_1, ..., f_p$, where $f_1^{p-1}$ is the validated prefix and $f_p$ is the word that the user types to correct the error. The following equation adds this feedback into Equation (1), with two constraints that apply to the range of words $1 \le i < p$:

$$\hat{y}_1^{\hat{I}} = \arg\max_{I, y_1^I} \Pr(y_1^I \mid x_1^J, \bar{y}_1^{\bar{I}}, f_1^p) = \arg\max_{I, y_1^I} \prod_{i=1}^{I} \Pr(y_i \mid y_1^{i-1}, x_1^J, \bar{y}_1^{\bar{I}}, f_1^p),$$

$$\text{subject to} \quad 1 \leq i < p,$$
$$f_i = y_i = \bar{y}_i,$$
$$f_p = y_p \neq \bar{y}_p,$$

(2)

where $\bar{y}_1^{\bar{I}} = \bar{y}_1, ..., \bar{y}_{\bar{I}}$ is the previous translation; $f_1^p$ is the feedback provided by the user, which corresponds the validated prefix with the new word typed; and $p$ is the length of the feedback. With the constraint $f_i = y_i = \bar{y}_i$, we ensure that all of the words before the error position (i.e., the validated prefix) remain in the new translation and, with the constraint $f_p = y_p \neq \bar{y}_p$, we force the use of the new word typed by the user. As the user corrects and validates the translation from left to right, in a more general way, this equation generates the most probable suffix for the provided prefix.

Equation (2) forces the system to use the word that the user has typed to fix the error; however, the user does not type the entire word when working at the character level. At this level, the system has to try to correct the translation using only a subset of characters from the correct word. As we no longer have the last word in full in the feedback, we need to make a few changes to Equation (2) in order to determine the highest probability word that starts with the same set of characters that the user has introduced:

$$\hat{y}_1^{\hat{I}} = \arg\max_{I, y_1^I} \Pr(y_1^I \mid x_1^J, \bar{y}_1^{\bar{I}}, f_1^p) = \arg\max_{I, y_1^I} \prod_{i=1}^{I} \Pr(y_i \mid y_1^{i-1}, x_1^J, \bar{y}_1^{\bar{I}}, f_1^p),$$

$$\text{subject to} \quad 1 \leq i < p,$$
$$f_i = y_i = \bar{y}_i,$$
$$y_p : \exists \hat{y}_p \hat{y}_{p+1}^{\hat{I}},$$

$$\hat{y}_p \hat{y}_{p+1}^{\hat{I}} = \arg\max_{\substack{I', y'_p, y'^I_{p+1} \\ f_{p\,1}^Q = y'^Q_{p\,1}}} \Pr(y'_p, y'^{I'}_{p+1} \mid x_1^J, y_1^{p-1}),$$

(3)

where $y_p$ is the word that the system is trying to correct, and $Q$ is the length of the last set of characters introduced. When we use the term $y_{p\,1}^Q$, we refer to characters 1 to $Q$ from the word at the $p^{\text{th}}$ position. We add a constraint to the suffix generation in Equation (3) to ensure that the system uses a valid word, according to the feedback entered by the user. The constraint $f_{p\,1}^Q = y'^Q_{p\,1}$ forces the system to find the highest probability suffix, in which the first word starts with the same characters as in the feedback $f_p$. Finally, $y_p \hat{y}_{p+1}^{\hat{I}}$ is the suffix with the highest probability, given the source sentence and the prefix validated by the user.

## 4. Mouse Actions

In conventional IPNMT systems, as described in the previous section, the user must use the keyboard and mouse to correct the translation; that is, they have to move the cursor to the error position with the mouse, and then type the correction. The correction may be the whole word or just a character, depending on whether the system works at the word or character level. We define a *mouse movement* as the action that occurs when the user changes the position of the cursor. We do not differentiate the distance the cursor moves; we always count the mouse movement as one. This is the user model that we adopt for our work, as it is the most common; however, some professional translators only use the keyboard during IPNMT sessions or, instead of a mouse, use a notepad for the selection of words.

We implement a more straightforward input method for IPNMT systems in this work. Instead of using the correct word that the user must type to fix the translation, we use the cursor position to solve the same problem. In this case, the model does not know what the

correct prediction looks like or what would have happened with a different prediction. The only information that the model has is that the next word or characters after the cursor are incorrect. This scenario is named *learning from bandit feedback* (This name is inherited from a model where a gambler pulls the arm of a different slot machine ('one-armed bandit') in each round, in order to maximize the final reward without knowing the optimal slot machine [30]).

This method removes the obligation of using a keyboard for the correction of translations. With just the error position obtained from the mouse, the system can generate a new translation that may fix the error. This action is named a Mouse Action (MA), and, depending on the context where it is used, we can differentiate between *non-explicit MAs* and *interactive-explicit MAs* [17].

*4.1. Non-Explicit MAs*

The main feature of non-explicit MAs is that they do not introduce an extra cost for the user. In conventional IPNMT systems, before the user types any character, they must move the cursor to the error position to execute the change. We use only this change in the mouse position to perform a non-explicit MA, as it provides enough information to generate a new translation. Knowing the first error position, we know that all of the previous words are correct, that the character next to the cursor is incorrect, and that we need to change it. The new translation generated by the system must have the same prefix and use a word with a different character in the error position. As the user must move the cursor to perform the correction in a conventional IPNMT system, this action does not incur extra costs, and the system performs the correction automatically when the mouse moves. This process does not ensure that the new suffix is correct, but the user behaves the same as in a conventional IPNMT system in the worst-case scenario.

Equation (3) uses the feedback provided by the user $f_1^p = f_1, ..., f_p$ (where $f_{p-1}^{\;Q}$ to $f_{p\;Q}$ are the last characters entered), in order to calculate the translation with the highest probability that fulfills the imposed constraints. In this new situation, the user does not need to type anything to perform the correction, but we know that the character at the error position must be different in the newly generated translation. In conventional IPNMT systems, the feedback includes the character typed by the user at $f_{p\;Q}$; however, now, $f_{p\;1}^{\;Q}$ are the characters before the cursor from the last word of the feedback. If the error is at the start of a word, $f_{p\;1}^{\;Q}$ remains empty. This situation can be expressed by adding a new constraint to the suffix generation method in Equation (3), as follows:

$$
\hat{y}_1^{\hat{I}} = \underset{I, y_1^I}{\arg\max} \ \Pr(y_1^I \mid x_1^J, \bar{y}_1^{\bar{I}}, f_1^p) = \underset{I, y_1^I}{\arg\max} \ \prod_{i=1}^{I} \Pr(y_i \mid y_1^{i-1}, x_1^J, \bar{y}_1^{\bar{I}}, f_1^p),
$$

$$
\text{subject to} \quad
\begin{aligned}
& 1 \le i < p, \\
& f_i = y_i = \bar{y}_i, \\
& y_p : \exists \, \hat{y}_p \hat{y}_{p+1}^{\hat{I}},
\end{aligned}
\tag{4}
$$

$$
\hat{y}_p \hat{y}_{p+1}^{\hat{I}} = \underset{\substack{I', y'_p, y'^{I'}_{p+1} \\ f_{p\;1}^{\;Q} = y'_{p\;1}^{\;Q} \\ y'_{p\;Q+1} \ne \bar{y}_{p\;Q+1}}}{\arg\max} \ \Pr(y'_p, y'^{I'}_{p+1} \mid x_1^J, y_1^{p-1}).
$$

To ensure that the new word at position $p$ does not have the same character as at position $Q + 1$, we include the constraint $y'_{p\;Q+1} \ne \bar{y}_{p\;Q+1}$ into Equation (3), in the part responsible for the generation of new suffixes. Figure 3 illustrates an IPNMT system, in which the user performs a non-explicit MA. At the start of the translation process, the system locates the mouse at the beginning of the translation. The user has not validated any part yet. The user moves the cursor to the first error in the first iteration, after the character '*v*' in the word '*improving*'. With this information, the system searches for a new word that starts with '*improv*' and does not follow with an '*i*'; in this case, the word '*improved*' is

determined. As the translation is correct in the second iteration, the user validates it and ends the translation process.

| | | |
|---|---|---|
| **SOURCE** (x): | | La situación actual debe mejorar sin duda alguna. |
| **REFERENCE** (y): | | The current situation must certainly be improved. |
| **ITER-0** | **(p)** | ( ) |
| | $(\hat{s}_h)$ | ‖ *The current situation must certainly be improving.* |
| **ITER-1** | **(p)** | The current situation must certainly be improv |
| | $(s_t)$ | ‖ *ing.* |
| | $(\hat{s}_h)$ | *ed.* |
| **ITER-2** | **( p)** | The current situation must certainly be improved. |
| | $(s_t)$ | ( ) |
| | $(k)$ | (#) |
| | $(\hat{s}_h)$ | ( ) |
| **FINAL** | $(p \equiv y)$ | The current situation must certainly be improved. |

**Figure 3.** Example of an IPNMT session with non-explicit MAs to translate a sentence from Spanish to English. Non-validated hypotheses are displayed in italics, and accepted prefixes are printed in normal font. The MAs are indicated by the symbol '‖'.

*4.2. Interactive-Explicit MAs*

Unlike non-explicit ones, interactive-explicit MAs suppose an extra effort cost for the user. As stated in the previous section, in a conventional IPNMT system, the user still has to move the cursor to the error position to fix it; therefore, we use this mouse movement to perform the non-explicit MAs without any extra cost. However, this action does not ensure that the error at that position will be fixed: the new word will have a different character at the error position, but it may still be incorrect. If the translation is not corrected after the non-explicit MA, the user can intentionally perform an interactive-explicit MA, as the mouse is already in position. For this action, as we already have the error position, we only need to send a signal to the system to perform a new MA at the same position. This signal can be sent with a mouse click, scrolling the mouse wheel, or by typing a special character such as `F1` or `Tab`. As the user must perform an action intentionally, this action introduces a little extra cost. The user has to decide which kind of action to execute, either performing an interactive-explicit MA or typing the correct character.

Each time that an interactive-explicit MA is performed at the same position, we obtain a new character that we do not want to appear in the new suffix that the system will generate. The following equation solves this problem by keeping track of the $k$ previous translations, where $k$ is the number of MAs performed in the same position:

$$\hat{y}_1^{\hat{I}} = \arg\max_{I, y_1^I} \Pr(y_1^I \mid x_1^J, \bar{y}_1^{\bar{I}}, f_1^p, k) = \arg\max_{I, y_1^I} \prod_{i=1}^{I} \Pr(y_i \mid y_1^{i-1}, x_1^J, \bar{y}_1^{\bar{I}}, f_1^p, k),$$

$$\text{subject to} \quad 1 \le i < p,$$

$$f_i = y_i = \bar{y}_i, \tag{5}$$

$$y_p : \exists \hat{y}_p^{(k)} \hat{y}_{p+1}^{\hat{I}},$$

$$\hat{y}_p^{(k)} \hat{y}_{p+1}^{\hat{I}} = \arg\max_{\substack{I', y'_p, y'^{I}_{p+1} \\ f_p{}_1^Q = y'_p{}_1^Q}} \Pr(y'_p, y'^{I'}_{p+1} \mid x_1^J, y_1^{p-1}),$$

$$y'_p{}_{Q+1} \notin \{\bar{y}_p{}_{Q+1}, y_p{}^{(1)}_{Q+1}, ..., y_p{}^{(k-1)}_{Q+1}\}$$

where $y_p^{(p)}$ is the word that occupies position $p$ of the new translation when the user has performed the $k^{\text{th}}$ MA; $y_p{}^{(l)}_{Q+1}$ $l < k$ is the set of characters at the error position for the previous mouse actions, which we do not want to appear in the starting character of the suffix; and $\bar{y}_p{}_{Q+1}$ is the character from the first translation, before performing any MA.

The set of characters is reset when the system fixes the character error and the user changes the cursor position.

Figure 4 illustrates an IPNMT system where the user performs both non-explicit and interactive-explicit MAs. The system locates the mouse at the beginning of the translation to start the validation and correction process. In the first iteration, the user moves the cursor to the middle of the word '*institution*' to correct it, performing a non-explicit MA. The system generates a new suffix that does not start with the character '*s*'. This new translation has not fixed the error and, so, the user now has to decide between performing an interactive-explicit MA or typing the correct character. In this case, the user decides to perform an interactive-explicit MA, and the system tries again to generate a new suffix, which cannot start with the characters '*s*' or '*v*'. This time, the system correctly generates the translation, and the user validates it by typing the special character '#', which ends the translation process.

| | | |
|---|---|---|
| **SOURCE** (x): | | Necesitamos una presentación mayor del ciudadano. |
| **REFERENCE** (y): | | We need a larger introduction from the citizens. |
| **ITER-0** | $(\mathbf{p})$ | ( ) |
| | $(\hat{s}_h)$ | ‖ *We need a larger institution for the citizens.* |
| **ITER-1** | $(\mathbf{p})$ | We need a larger in |
| | $(s_t)$ | ‖ *stitution for the citizens.* |
| | $(\hat{s}_h)$ | *vestiture of the citizens.* |
| **ITER-2** | $(\mathbf{p})$ | We need a larger in |
| | $(s_t)$ | ‖ *vestiture of the citizens.* |
| | $(\hat{s}_h)$ | *troduction from the citizens.* |
| **ITER-3** | $(\mathbf{p})$ | We need a larger introduction from the citizens. |
| | $(s_t)$ | ( ) |
| | $(k)$ | (#) |
| | $(\hat{s}_h)$ | ( ) |
| **FINAL** | $(p \equiv y)$ | We need a larger introduction from the citizens. |

**Figure 4.** Example of an IPNMT session with non-explicit and interactive-explicit MAs to translate a sentence from Spanish to English. Non-validated hypotheses are displayed in italics, and accepted prefixes are printed in normal font. The MAs are indicated by the symbol '‖'.

## 5. Experimental Setup

### 5.1. Corpora

For this work, we utilized the public corpus Europarl [31], which has been extracted from the Proceedings of the Europarl Parliament, is publicly available to everyone on the internet, and has been translated to all of the official languages of the European Union. We used the German–English (De-En), Spanish–English (Es-En), and French–English (Fr-En) language pairs, in both directions, in our experiments. The characteristics of each pair of languages from the corpus are described in Table 1. We used the Moses toolkit, developed by Koehn et al. (2007) [32], to clean, lower-case, and tokenize the corpus. Then, we applied the subword subdivision Byte Pair Encoding (BPE) technique, developed by Sennrich et al. (2006) [33], with a maximum of 32,000 merges. This technique segments the words in our vocabulary into smaller fragments, making the NMT model capable of encoding rare and unknown words into sequences of subword units.

**Table 1.** Characteristics of the Europarl corpus for the German–English (De-En), Spanish–English (Es-En), and French–English (Fr-En) language pairs. *K* and *M* denote thousands and millions, respectively.

|  |  | De-En | | Es-En | | Fr-En | |
|---|---|---|---|---|---|---|---|
| Training | Sentences | 751K | | 730K | | 688K | |
| | Avg. Length | 20 | 21 | 21 | 20 | 22 | 20 |
| | Run. Words | 15M | 16M | 15M | 15M | 15M | 14M |
| | Vocabulary | 195K | 65K | 102K | 64K | 80K | 61K |
| Development | Sentences | 2000 | | 2000 | | 2000 | |
| | Avg. Length | 27 | 29 | 30 | 29 | 33 | 29 |
| | Run. Words | 55K | 59K | 60K | 59K | 67K | 59K |
| Test | Sentences | 2000 | | 2000 | | 2000 | |
| | Avg. Length | 27 | 29 | 30 | 29 | 33 | 29 |
| | Run. Words | 54K | 58K | 67K | 58K | 66K | 58K |

## 5.2. Model Architecture

We built our NMT models using the open-source toolkit NMT-Keras [34]. Our models follow the Transformer architecture [35], with a word embedding and dimension size of 512. We set the hidden and output dimensions of the output layer to 2048 and 512, respectively. Each multi-head attention layer had eight heads, and we stacked six encoder and decoder layers. The models used Adam [36] as the learning algorithm, with a learning rate of 0.0002. We clipped the $L_2$ norm of the gradient to 5, set the batch size to 30, and set the beam size to 6. Table 2 shows the translation performance, in terms of the BiLingual Evaluation Understudy (Bleu) of the transformers trained in the Europarl task.

**Table 2.** Translation quality of our NMT models for the Europarl task, in terms of BLEU.

|  | BLEU($\uparrow$) |
|---|---|
| De-En | 28.8 |
| En-De | 19.2 |
| Es-En | 32.1 |
| En-Es | 31.4 |
| Fr-En | 31.1 |
| En-Fr | 32.3 |

## 5.3. System Evaluation

In this work, we consider two kinds of actions that we want to compare: The use of a keyboard to correct the translations, and the use of mouse actions. Both suppose an effort for the user; however, with the assumption that mouse actions suppose a lower effort, we wished to compare how many keystrokes the user can save through the use of our method. To quantify the effort required by the user when typing the correct characters, we used the Character Stroke Ratio (CSR) metric. In Section 5.5, we will compare our CSR reduction with the Word Stroke Ratio (WSR) reduction reported in a previous work [18], as both represent the human effort performed. We also used metrics to account for the mouse actions, including the character Mouse Action Ratio (cMAR) and the useful Mouse Action Ratio (uMAR).

The CSR [37] is computed as the number of characters that the user must type during an IPNMT session to translate all of the sentences correctly, normalized with respect to the total number of characters. This metric provides the percentage of characters that the user has entered in the translation session.

The WSR [38] is very similar to CSR, but is used at the word level. It is computed as the number of words that the user needs to type to correct all the translations, normalized by the total number of words. This metric provides the percentage of words that the user has entered in the translation session.

The cMAR [14] is computed as the number of MAs that the user performs throughout the translation session, normalized by the total number of characters. This metric also provides us with the percentage of MAs performed, compared with the total number of characters; however, in this case, the metric can exceed 100% if we allow the user to use interactive-explicit MAs, as they can perform multiple MAs per character.

The uMAR [14] indicates the amount of MAs that are required to achieve the translation that the user has in mind (i.e., the MAs that actually result in changing the first character of the suffix correctly). Formally, the uMAR is defined as follows:

$$uMAR = \frac{MAC - n\,CSC}{MAC}, \tag{6}$$

where the Mouse Action Count (MAC) is the total number of MAs performed, the Character Stroke Count (CSC) is the number of characters typed, and $n$ is the maximum amount of MA allowed before the user types in a character. Note that, in order to perform a keystroke, the user previously must have performed $n$ MAs; therefore, in Equation (6), we are removing from the total count of MAs those that were not useful and did not help to find the correct character.

*5.4. User Simulation*

Using professional translators to perform experiments with an IPNMT system is very costly and slow. Furthermore, their use implies external factors that could interfere with the results of our experiments, such as the cognitive cost of thinking about how to correct an error. For this reason, we simulated the expected behavior of a professional translator in our IPNMT system.

First, we consider the conventional behavior, where the user does not perform any MA and always types the correct character. We compare the translation generated by the system with the reference one, in order to determine the location of the first error. The simulated user looks at the reference to find which character should be in the error position, and then types it. The system uses this feedback to generate a new translation. We repeat this process until the generated translation is equal to the reference. Then, the simulated user introduces the validation token, and the process finishes.

Second, we expanded upon the previous behavior by using non-explicit MAs. Now, when the position of the first error is determined from the translation by looking at the reference, the simulated user performs an MA, which is sent as error position feedback to the system. The system uses this information to generate a new translation, which may fix the error. This is compared with the translation reference, to figure out if the error persists. If this is the case, the simulated user follows the conventional behavior, and types the correct character. This process is repeated until the generated translation matches the reference, at which point the simulated user introduces the validation token.

Finally, in the last behavior mode, we added interactive-explicit MAs. In this case, we start the experiments by setting a maximum number of MAs to perform per error (this number must be greater than 1). As in the previous case, when the position of the first error is determined, the simulated user sends it to the system. The main difference is that, if the system does not correct the error properly, another MA is performed, until the set maximum number is achieved. If the system has not corrected the translation at the error position when the maximum number of MAs is reached, then we proceed as in the conventional system and type the correct character. The process is repeated until the generated translation matches the reference, and the validation token is introduced.

The behavior used in the IPNMT system depends on the maximum number of MAs: namely, for the conventional behavior, we set a maximum of zero; for the non-explicit behavior, we set a maximum of one; and for the interactive-explicit, we set a maximum number of MAs greater than one. The user model that we simulated with different behaviors has previously been used in previous works considering SMT and NMT models [14,17,18].

*5.5. Results*

For our experiments, we carried out an IPNMT session for each pair of languages in the Europarl corpus. We repeated the experiment for each pair of languages with a different number of maximum MAs. We started with zero, where the system behaved as a conventional IPNMT system, then with only one MA, where we only considered non-explicit MAs, up to a maximum of five MAs per error position. The results obtained from the experiments are given in Table 3.

We first compared the results obtained with a conventional IPNMT system to those obtained with the addition of non-explicit MAs and interactive-explicit MAs. For each of the cases, we display the effort performed by the user, in terms of cMAR and CSR. In the cases where MAs were performed, we include *CSR relative* metric, which describes how much the required effort was reduced, when compared with the baseline. In the baseline case, looking at the cMAR obtained, the user needed to type 15% of the characters from the translations, on average. When we added non-explicit MAs to the system, the effort required by the user reduced by about half. Such actions do not suppose an extra cost for the user, as the mouse movement required to perform the mouse action is also performed in a conventional IPNMT session. For this reason, the cMAR did not change from the baseline to the non-explicit case. When we increased the maximum number of MAs that the user could perform at each error position to five, the reduction in human effort increased to 80%.

**Table 3.** Experimental results in the Europarl corpus when considering non-explicit and interactive-explicit MAs.

|  | Baseline | | Non-Explicit | | | Interaction-Explicit | | |
|---|---|---|---|---|---|---|---|---|
|  | cMAR ($\downarrow$) | CSR ($\downarrow$) | cMAR ($\downarrow$) | CSR ($\downarrow$) | CSR rel. ($\uparrow$) | cMAR ($\downarrow$) | CSR ($\downarrow$) | CSR rel. ($\uparrow$) |
| De-En | 14.70 | 15.92 | 14.70 | 7.06 | 55.65 | 30.91 | 2.61 | 83.61 |
| En-De | 16.53 | 17.56 | 16.53 | 8.24 | 53.08 | 36.76 | 3.45 | 80.35 |
| Es-En | 14.01 | 15.19 | 14.01 | 6.68 | 56.02 | 29.31 | 2.43 | 84.00 |
| En-Es | 14.15 | 15.16 | 14.15 | 6.60 | 56.46 | 29.19 | 2.38 | 84.30 |
| Fr-En | 14.23 | 15.44 | 14.23 | 6.71 | 56.54 | 29.51 | 2.46 | 84.07 |
| En-Fr | 13.10 | 13.93 | 13.10 | 6.08 | 56.35 | 27.15 | 2.25 | 83.85 |

This table demonstrates the significant utility of MAs. Without supposing an extra cost for the user, they already reduced the number of characters that the user had to type by half. Table 4 presents the results of Navarro and Casacuberta (2021) [18], where the MAs were implemented at the word level. The results are shown in terms of the metric WSR, which describes the human effort with respect to the ratio of words typed. With only non-explicit MAs, they obtained a reduction of 28%. This method has, therefore, shown its usefulness in both techniques, while obtaining a higher reduction at the character level. In the article of Sanchis-Trilles et al. (2008) [17], the MAs were implemented at the word level in an IPSMT system. In this case, with non-explicit MAs, they obtained a WSR reduction of 6%. Considering these results, we can see how the performance has improved along with advances in MT models.

**Table 4.** Experimental results in the Europarl corpus when considering non-explicit and interactive-explicit MAs at word level. Navarro et al. (2021) [18].

|  | Baseline | | Non-Explicit | | | Interaction-Explicit | | |
|---|---|---|---|---|---|---|---|---|
|  | MAR ($\downarrow$) | WSR ($\downarrow$) | MAR ($\downarrow$) | WSR ($\downarrow$) | WSR rel. ($\uparrow$) | MAR ($\downarrow$) | WSR ($\downarrow$) | WSR rel. ($\uparrow$) |
| De-En | 42.5 | 40.5 | 44.3 | 29.1 | 28.2 | 136.7 | 17.5 | 56.7 |
| En-De | 49.7 | 47.8 | 51.8 | 36.2 | 24.3 | 173.1 | 24.5 | 48.8 |
| Es-En | 40.5 | 38.2 | 42.2 | 27.0 | 29.3 | 127.9 | 16.3 | 57.4 |
| En-Es | 41.4 | 39.6 | 43.3 | 28.7 | 27.6 | 135.9 | 17.8 | 55.1 |
| Fr-En | 41.2 | 38.9 | 42.9 | 27.3 | 29.9 | 129.6 | 16.4 | 58.0 |
| En-Fr | 38.1 | 36.2 | 39.7 | 25.7 | 29.0 | 121.2 | 15.3 | 57.7 |

Figure 5 shows the CSR and MAR scores for each maximum number of MAs tested. These graphs show the reliability and consistency of the proposed method. We obtained similar results for each pair of languages and, so, we consider that the results obtained may be very similar in new corpora and languages pairs. This fact is very positive as, with only the first MA, the human effort was significantly reduced, and each of the following mouse actions further increased this reduction; however, with each increase, the effort reduction is lower. However, if we wish to know how much effort we can reduce with each of the mouse actions, we must look at the uMAR scores, which give us the percentage of useful MAs for each of the maximum values used.

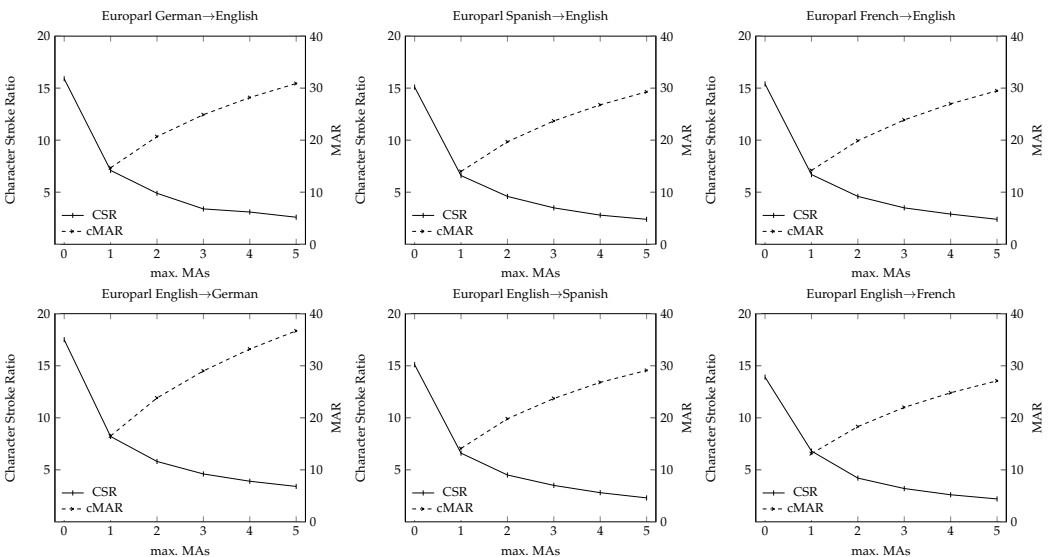

**Figure 5.** CSR and cMAR, when considering up to five maximum MAs in the Europarl corpus.

Figure 6 displays the uMAR scores when considering one up to five maximum MAs. For each step in the graph, the uMAR score tells us the percentage of useful MAs, and which ones have ended correcting the error. From these results, we can understand why the required effort was significantly reduced with the non-explicit mouse actions. As we had an uMAR of 50, approximately 50% of the mouse actions performed fixed the error correctly. We should highlight that the uMAR score did not decrease for each MA, remaining at 50% and even increasing to 60%; this means that, each time the user performs an MA, the system had a 50% probability of fixing the error correctly. If we again look at the previous work [18], their uMAR score for the same corpus was near 30%, while Sanchis et al. (2008) [17] obtained a 15% uMAR. In these cases, the probability of fixing the error at each attempt was lower and, for this reason, our method obtained a higher reduction than the others when we used up to five maximum MAs. Using the MAs at the character level obtained better results, leading to a higher percentage of useful MAs. This information is beneficial for the user as, in the end, they must decide at each moment whether perform an MA or to directly type the correct word. In our case, each of the performed mouse actions had a 50% possibility of fixing the error correctly.

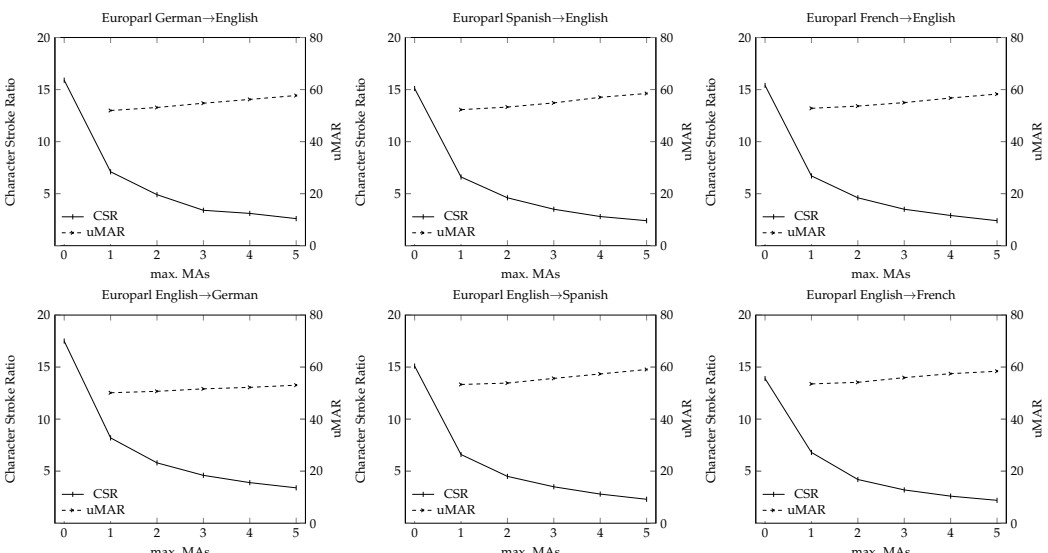

**Figure 6.** CSR and uMAR when considering up to five maximum MAs in the Europarl corpus.

## 6. Discussion

In this article, we implemented the use of bandit feedback, in terms of MAs, to generate new translations without the professional user needing to type anything. We tested this method in three different languages pairs from the Europarl corpus, by carrying out IPNMT sessions with a maximum of 0 up to 5 MAs per error position. With these experiments, we determined how human effort can be reduced by using this method, as well as the helpfulness of each mouse action performed.

The effort reduction that we achieved was very significant. When only performing non-explicit MAs—which does not introduce an extra cost for the user—the human effort was reduced by 50%. When we used a maximum of 5 MAs per error position, this reduction was increased to 80%. These reductions are higher than those obtained when this same method was used at the word level. As such, we demonstrated the effectiveness of the proposed method in IPNMT systems, and there is no reason not to implement it in these systems; at least the non-explicit MAs, which do not lead to any extra cost.

Additionally, for the interactive-explicit MAs, where the user has to decide between performing them or typing the correct character, we considered a metric that indicates the probability of the system correcting the error per each MA. With this score, the user can support their decision with a value. This score increased with each MA included in our experiments, making it more probable that the system fixes the error each time.

These results demonstrate that MAs can be very useful for the IPNMT sessions in a simulated environment, significantly reducing the effort required of the user during translation sessions. In the future, we intend to perform these same experiments with real translators, where we do not have a reference, in order to validate the results obtained in this paper.

**Author Contributions:** Conceptualization Á.N. and F.C.; formal analysis Á.N. and F.C.; investigation Á.N.; methodology Á.N.; software Á.N.; supervision F.C.; validation Á.N.; visualization Á.N.; writing—original draft Á.N.; writing—review and editing Á.N. and F.C. All authors have read and agreed to the published version of the manuscript.

**Funding:** This work received funds from the Comunitat Valenciana under project EU-FEDER (*ID-IFEDER*/2018/025), Generalitat Valenciana under project ALMAMATER (*PrometeoII*/2014/030), and Ministerio de Ciencia e Investigación/Agencia Estatal de Investigacion /10.13039/501100011033/, "FEDER Una manera de hacer Europa" under project MIRANDA-DocTIUM (RTI2018-095645-B-C22), and Universitat Politècnica de València under the program (PAID-01-21).

**Informed Consent Statement:** Not applicable.

**Data Availability Statement:** Not applicable.

**Conflicts of Interest:** The authors declare no conflict of interest.

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
