# Peer review of "On the Use of Mouse Actions at the Character Level"

_information, doi:10.3390/info13060294_

Round 1
Reviewer 1 Report
The paper is well-written and the topic is interesting. However, the authors need to improve the paper's quality based on the following comments so that the paper gets all qualifications factors for publication. I emphasize that the current version of the paper is not qualified for publication yet.
1) The Abstract needs to be improved. It is now sloppy and the authors need to elaborate on it by emphasizing more about the proposed method.
2) The problem statement(s) and proposed solution need to be clearly mentioned in the section Introduction.
3) A section entitled "Related Work" needs to be added to the revised version of the paper. It can take place after the Introduction section or before the conclusions.
4) The following references need to be added to the related work section:
4-1) Augmenting Neural Machine Translation through Round-Trip Training Approach. (Ahmadnia and Dorr, 2019). Open Computer Science (De Gruyter). 9(1):268-278.
4-2) Dual Learning for Machine Translation. (He et al., 2016). Neural Information Processing Systems.
4-3) Fully Character-level Neural Machine Translation without Explicit Segmentation. (Lee et al., 2017). Transactions of the Associations for Computational Linguistics.
5) The paper suffers from the lack of detailed analysis in Fig.6.
6) Results comparison to the state-of-the-art is highly recommended.
7) Proofreading for submitting the revised version is recommended.
Author Response
1) I have better described the bandit feedback that obtains the system to generate the translations and how the user can use it at the error positions multiple times to correct the translations without typing anything, reducing the keystrokes performed during the translation session.
2) I have described better at the Introduction which is the problem that we try to solve and the technique that we use.
3) I have added the Section Related Work
4) I have added the Literature Review section. I have read the three papers referenced to add. Still, none of them is related to the interactive field, which is our paper's main line of investigation.
5) I have expanded the analysis performed in figure 6.
6) In the Related Work section, I have mentioned the start-of-art and added a better comparison of results.
Reviewer 2 Report
The paper proposes a method for reducing the effort of correcting MT based on Mouse Actions. It seems that the method produces good results, however, neither the method nor the state-of-the-art in this field is properly described, which makes judging the actual contribution very difficult.
Detailed comments:
Section 1: three things need to be separated. First, the introduction - what is the problem and how its solutions are evaluated, i.e. what are the metrics. Second, literature review - what are the main method classes, and what is the current SOTA. Third, what is suggested in this paper and how does it advance the knowledge. Additionally, what type of translation is studied here? What language(s)? What type(s) of texts? General? News articles? Scientific papers?
All of this is currently missing from the paper.
l.131 'bandit feedback' is undefined.
l. 145 What is cursor movement? What is considered to be 1 movement? Using a mouse means jumping to a different location, how is it counted?
l.170 This sentence is grammatically incorrect. What does it mean? Please use professional editing services to edit your paper.
l.213 Abbreviations, such as BPE, have to be defined.
l.231 What previous work? Why this one? Why not several?
Section 4.4 is completely unclear. If the user is 'simulated', then it is an automatic system. How is it implemented? What's the algorithm? Is it the same for all languages addressed in the task?
Table 3: Why is your method compared to one baseline only? Where are the competing systems? These systems need to be (a) described in the proper Related Work section, and (b) compared to in the Experiments section.
Author Response
In the last paragraph of the introduction, I have added a brief description of the problem, our solution, and the metric we are reducing. I have also added a Related Work section with the SOTA and explained how our work advances the knowledge.
l.131 -> I have added to the added which is the information that we use as bandit feedback. And at this line, I have added the description of what bandit feedback is.
l.145 -> I have described what we count as mouse movement at the start of section 3, where we define the user model used.
l.170 -> I think I have corrected the sentence. The meaning of the phrase is explained in the following, where I briefly repeat why non-explicit MAs do not imply an extra cost for the user, and I expand it with the use and definition of interactive-explicit MAs.
l.213 -> I have added a brief explanation of the BPE technique with the abbreviation definition.
Section 4.4 -> I have rewritten Section 4.4, explaining in more detail the simulation method.
Round 2
Reviewer 2 Report
My comments were adequately addressed. I think this manuscript can be accepted because the results are good enough and are above SOTA.